# Co-transcriptional R-loops are the main cause of estrogen-induced DNA damage

Caroline Townsend Stork[1], Michael Bocek[1], Madzia P Crossley[1], Julie Sollier[1], Lionel A Sanz[2], Frédéric Chédin[2], Tomek Swigut[1], Karlene A Cimprich[1]*

[1]Department of Chemical and Systems Biology, Stanford University School of Medicine, Stanford, United States; [2]Department of Molecular and Cellular Biology, University of California, Davis, Davis, United States

**Abstract** The hormone estrogen (E2) binds the estrogen receptor to promote transcription of E2-responsive genes in the breast and other tissues. E2 also has links to genomic instability, and elevated E2 levels are tied to breast cancer. Here, we show that E2 stimulation causes a rapid, global increase in the formation of R-loops, co-transcriptional RNA-DNA products, which in some instances have been linked to DNA damage. We show that E2-dependent R-loop formation and breast cancer rearrangements are highly enriched at E2-responsive genomic loci and that E2 induces DNA replication-dependent double-strand breaks (DSBs). Strikingly, many DSBs that accumulate in response to E2 are R-loop dependent. Thus, R-loops resulting from the E2 transcriptional response are a significant source of DNA damage. This work reveals a novel mechanism by which E2 stimulation leads to genomic instability and highlights how transcriptional programs play an important role in shaping the genomic landscape of DNA damage susceptibility.

## Introduction

The hormone estrogen (E2, 17β-estradiol) is essential for the development and function of mammary tissue (*Bieche et al., 2001*), stimulating a transcriptional program that drives breast cell proliferation. Paradoxically, E2 exposure is also associated with an elevated risk of breast carcinogenesis (*Liehr, 2000*; *Yager and Davidson, 2006*). Specifically, higher E2 serum concentrations and longer lifetime E2 exposure are both positively correlated with an increased incidence of sporadic breast cancer (*Clemons and Goss, 2001*; *Colditz, 1998*; *Hilakivi-Clarke et al., 2002*). Breast cancers exhibit a large number of chromosomal abnormalities, including mutations and copy number alterations (*Nik-Zainal et al., 2016*). Moreover, E2 leads to DNA damage in breast epithelial cells that express the estrogen receptor (ER) (*Liehr, 2000*; *Williamson and Lees-Miller, 2011*), and in rat models, E2 stimulation is causally linked to chromosome instability and aneuploidy (*Li et al., 2004*). Despite strong links between estrogen and genomic instability, the molecular mechanism by which E2 causes this instability in breast cancer is unclear.

Functionally, E2 is a key transcriptional regulator that governs the expression of thousands of genes in breast cells (*Cheung and Kraus, 2010*; *Hah et al., 2011*) through its association with the nuclear hormone receptors ER-α and ER-β. Upon translocation into the nucleus, the E2-ER complex binds to estrogen-response elements or to other transcription factors, thus altering gene expression (*Marino et al., 2006*). Among the genes induced by E2 are a number that are important for cell proliferation (*Gong et al., 2014*). Thus, one proposed model to explain E2-induced genome instability is that the uncontrolled proliferation driven by deregulation of genes such as Cyclin D1 causes replication stress and DNA damage (*Caldon, 2014*; *Halazonetis et al., 2008*). However, another relatively unexplored hypothesis is that the dramatic increase in transcription itself contributes to E2-induced DNA damage.

*For correspondence: cimprich@stanford.edu

**Competing interests:** The authors declare that no competing interests exist.

**eLife digest** The hormone estrogen controls the development of breast tissue. However too much estrogen can damage the DNA in human cells and may be linked to an increased risk of breast cancer. In breast cells, estrogen activates many genes via a process called transcription. The transcription process results in the production of an RNA molecule that contains a copy of the instructions encoded within the gene.

Previous studies have found that, in certain cases, a new RNA molecule can stick to the matching DNA from which it was made. This creates a structure known as an R-loop, which can lead the DNA to break. DNA breaks are particularly harmful because they can dramatically alter the cell's genome in ways that allow it to become cancerous. However, it was not clear if the large increase in transcription triggered by estrogen causes an increase in R-loops, which could help to explain the DNA damage that has been reported to occur when cells are treated with estrogen.

Now, Stork et al. show that treating human breast cancer cells with estrogen causes an increase in R-loops and DNA breaks. The R-loops occurred particularly in regions of the genome that contain estrogen-activated genes. Stork et al. also found that regions of estrogen-activated transcription were more frequently mutated in breast cancers, and further experiments confirmed that the R-loops were responsible for many of the DNA breaks that occurred following estrogen treatment. Taken together, these findings demonstrate that the changes in transcription due to estrogen lead to increased R-loops and DNA breaks, which may make the cells vulnerable to becoming cancerous.

The next challenge is to determine precisely where these DNA breaks that result from estrogen occur on the DNA. Knowing the location of the DNA breaks will be useful in determining what additional factors or genomic features make an R-loop more prone to being broken. This in turn might help explain how the R-loops lead to DNA damage. In addition, further studies are also needed to determine if tumor samples from breast cancer patients also contain increased levels of R-loops.

Co-transcriptional structures known as R-loops form upon hybridization of nascent RNA with the template DNA strand and are prevalent in mammalian genomes. These structures are proposed to serve regulatory roles in the cell including the patterning of promoter chromatin and the facilitation of transcription termination (*Ginno et al., 2012*; *Skourti-Stathaki et al., 2011*). In certain contexts, however, R-loops may serve as precursors for DSBs. The coordinated cleavage of an R-loop at the switch region in B-cells facilitates recombination and generates antibody diversity (*Santos-Pereira and Aguilera, 2015*; *Yu et al., 2003*). Changes in R-loop levels, moreover, are clearly associated with increased DNA damage and genomic instability. For example, the loss of RNA processing and R-loop regulatory factors results in an increase in R-loops and R-loop-dependent DNA damage in both yeast and human cells (*Garcia-Rubio et al., 2015*; *Huertas and Aguilera, 2003*; *Li and Manley, 2005*; *Paulsen et al., 2009*; *Sollier et al., 2014*; *Tuduri et al., 2009*). Additionally, highly transcribed loci may be more prone to R-loop formation. This is evidenced by the enrichment of R-loops at rDNA repeats (*El Hage et al., 2010*) and the binding of the R-loop suppressing factor Npl3 preferentially at the most highly transcribed genes (*Santos-Pereira et al., 2013*).

While many studies have shown that the absence of RNA processing factors can lead to R-loop-dependent DNA damage, little is known about how more physiological changes, such as those induced by hormones or environmental stimuli, may influence R-loop formation and DNA damage in human cells. Dramatic and rapid changes in gene expression could challenge co-transcriptional processing pathways resulting in R-loop formation. Here, we tested whether the formation of R-loops following increased gene expression is a source of E2-induced DNA damage in ER-positive breast epithelial cells. We report that E2 treatment leads to a dramatic increase in R-loops, most strongly at E2-responsive genes. E2-responsive genes, moreover, are enriched in breast cancer rearrangements. We find that E2 treatment also leads to a significant increase in DNA replication- and transcription-dependent DNA damage signaling and DSBs through an R-loop dependent process. Collectively, our data indicate that physiological changes in transcription can drive R-loop formation to promote genomic instability in human cells.

## Results

### E2 induces transcription and replication-dependent DSBs

We first sought to characterize the DNA damage that arises in response to E2 stimulation in ER-positive breast epithelial cells (*Williamson and Lees-Miller, 2011*). To do so, we examined phosphorylation of the histone variant H2AX (P-H2AX), a marker of DNA damage, after treatment of hormone-starved MCF7 cells with different doses of E2. We observed a clear, dose-dependent increase in P-H2AX intensity 24 hr after E2 addition (*Figure 1A,B*). E2 treatment also resulted in an increase in 53BP1 foci (*Figure 1—figure supplement 1*), another marker of damaged DNA. Hormone-free culture conditions cause ER-positive cells to withdraw from the cell cycle, and low P-H2AX levels arise during normal DNA replication (*Mirzoeva and Petrini, 2003*). Thus, we asked whether an increase in the percentage of cells in S phase following E2 stimulation could account for the increased P-H2AX signal. Notably, cells treated with 1 or 100 nM E2 had similar cell cycle profiles (*Figure 1—figure supplement 2*), despite significantly different P-H2AX levels, suggesting that the increase in DNA damage was not due to differences in the S phase population. Importantly, ER-negative MCF10A breast epithelial cells showed no significant change in P-H2AX signal when treated with 100 nM E2 (*Figure 1—figure supplements 3A,B*), consistent with the previous finding that E2-induced DNA damage depends on the ER (*Williamson and Lees-Miller, 2011*). We also asked if E2 treatment leads to an increase in DSBs or if the observed changes in DNA damage response (DDR) signaling are induced by some alternative mechanism. The neutral comet assay provided direct evidence that E2 leads to DSB formation (*Figure 1C,D*).

Previous studies have shown that E2 induces the rapid and transient formation of DNA breaks necessary for transcriptional induction at E2 promoters, and these breaks have been reported to be dependent upon the activity of TopoIIβ and APOBEC3B (*Ju et al., 2006*; *Periyasamy et al., 2015*). When we examined this early response to E2, we found that the P-H2AX signal resulting from 24 hr of E2 stimulation was significantly greater than the signal resulting from very short (10–40 min) treatments, raising the possibility that the later DNA breaks arise through a different mechanism (*Figure 1E*). Given that a significant portion of cells would have entered S-phase by the 24-hr time point, we asked whether DNA replication was associated with E2-induced damage by pulsing cells with EdU at various times after E2 addition and monitoring H2AX phosphorylation. The percentage of P-H2AX positive cells increased dramatically after 16 and 24 hr, and the majority of these cells were EdU positive (*Figure 1 F,G*, and *Figure 1—figure supplement 4*). This finding strongly suggests that E2-induced DNA damage may require DNA replication. Indeed, when we treated E2-stimulated cells with a Cdc7 inhibitor (PHA 767491) that largely blocks S phase entry (*Figure 1—figure supplement 5A*), we found that the DNA damage was reduced (*Figure 1H*). At the 1 µM dose of PHA 767491 used, phosphorylation of RNA Pol II at Ser2 by Cdk9 was not inhibited, indicating the effect on P-H2AX is not a result of inhibiting RNA Pol II elongation (*Figure 1—figure supplement 5B*). We also asked if the late phase of E2-induced DNA damage is affected by blocking transcription coincident with S-phase entry. Treatment with the transcription inhibitor flavopiridol abolished the late E2-induced increase in P-H2AX (*Figure 1I*) but did not significantly affect the percentage of cells in S phase (*Figure 1—figure supplement 6*). Taken together, these results demonstrate that E2 induces DSBs in a replication- and transcription-dependent manner.

### Estrogen induces robust R-loop formation at E2-responsive genes

E2 stimulation causes a rapid burst of transcription at a large number of genes (*Hah et al., 2011*). As defects in co-transcriptional processing have been shown to promote R-loop formation (*Hamperl and Cimprich, 2014*; *Santos-Pereira and Aguilera, 2015*; *Sollier and Cimprich, 2015*), we asked whether E2 stimulation, which might saturate co-transcriptional processing pathways, leads to elevated R-loop levels. To do so, we probed genomic DNA from mock- or E2-treated cells (24 hr) with the anti RNA-DNA hybrid monoclonal antibody S9.6 (*Boguslawski et al., 1986*). We observed a dramatic increase in RNA-DNA hybrids upon E2 treatment (*Figure 2A*). Moreover, RNase H treatment, which specifically removes the RNA strand of an RNA-DNA hybrid, abolished the signal (*Figure 2A*), indicating antibody specificity for RNA-DNA hybrids.

To directly determine the location and abundance of R-loops genome-wide, we performed DNA-RNA immunoprecipitation (DRIP) with this antibody followed by next generation sequencing on cells

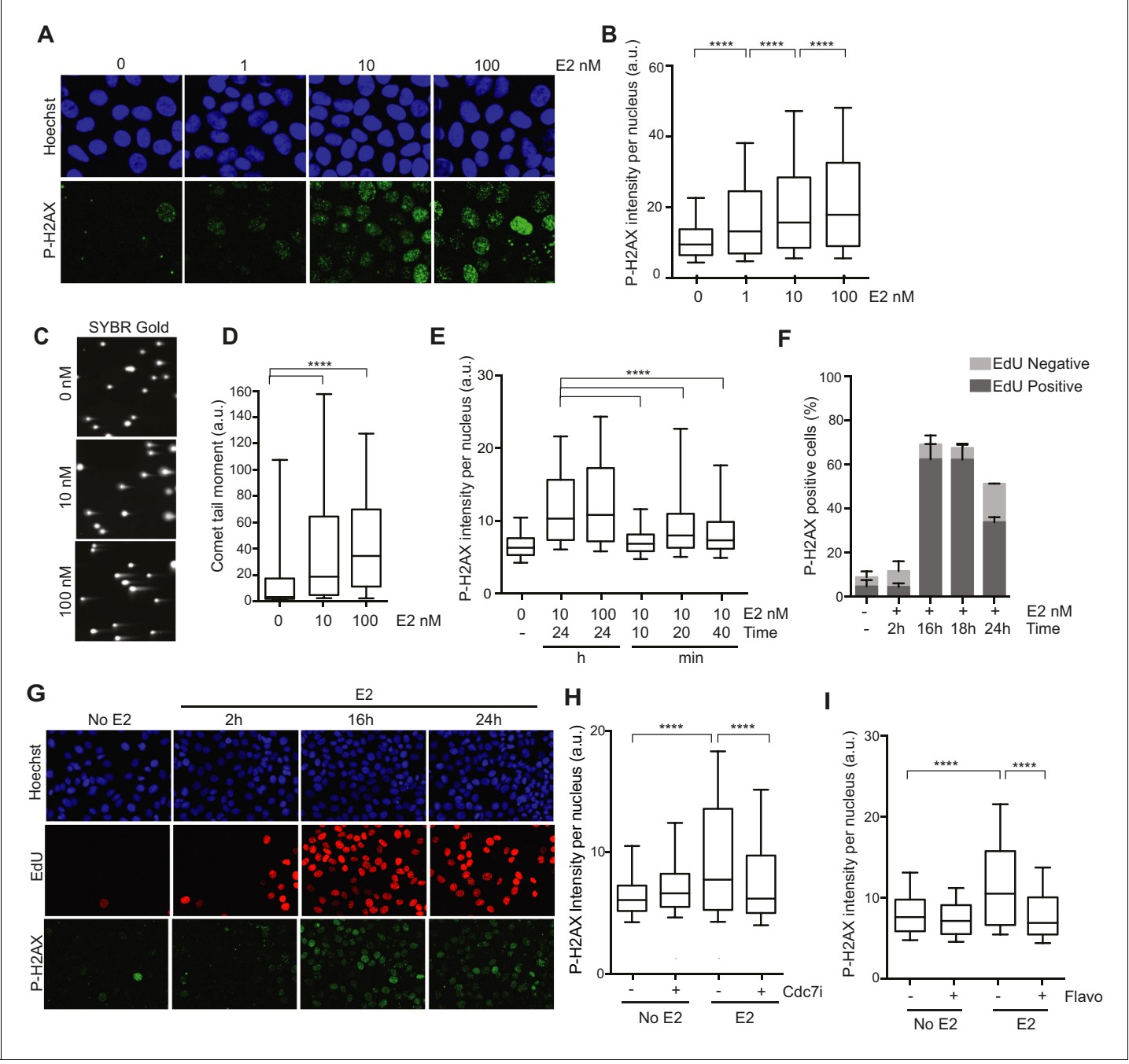

**Figure 1.** Estrogen induces DNA damage and DSBs in a replication-dependent manner. (**A**) Immunostaining for P-H2AX in MCF7 cells treated with 0, 1, 10, or 100 nM E2 for 24 hr. (**B**) Quantification of P-H2AX immunostaining for data shown in (**A**), ****p<0.0001. n = 2 biological replicates. (**C**) Neutral comet assay in MCF7 cells treated with 0, 10, or 100 nM E2 for 24 hr. (**D**) Quantification of the neutral comet tail moment for data in (**C**). ****p<0.0001. n = 4 biological replicates. ≥50 comets/condition. (**E**) Quantification of P-H2AX immunostaining per nucleus in cells treated with 0, 10 or 100 nM E2 for indicated time prior to fixation. min = minutes, h = hours. ****p<0.0001. n = 3 biological replicates. (**F**) Quantification of the percent of P-H2AX positive cells and EdU staining in cells treated with 0 or 100 nM E2 for the indicated time. Cells were pulsed for 30 min with 10 μM EdU prior to fixation. Error bars represent SD of 2 biological replicates. (**G**) Immunostaining for EdU and P-H2AX for the experiment described in (**F**). (**H**) Quantification of P-H2AX immunostaining per nucleus in cells treated with 0 or 100 nM E2 concurrently with DMSO or 1 μM Cdc7 inhibitor PHA 767491 for 14 hr. Cells were pulsed with 10 μM EdU 30 min prior to fixation. n = 3 biological replicates. (**I**) Quantification of P-H2AX immunostaining in MCF7 cells treated with 0 or 100 nM E2 for 12 hr prior to the addition of 0.8 μM flavopiridol or DMSO for 2 hr. Cells were pulsed with 10 μM EdU for 30 min prior to harvesting. ****p<0.0001. n = 3 biological replicates. For all graphs: box and whiskers represent 25–75 and 10–90 percentiles, respectively. The line represents the median value. a.u. = arbitrary units. Associated p-values are from non-parametric Mann-Whitney rank sum t-test. >1000 cells/condition unless noted.

*Figure 1 continued on next page*

*Figure 1 continued*

The following figure supplements are available for figure 1:

**Figure supplement 1.** Immunostaining for 53BP1 in MCF7 cells either treated with 0 or 100 nM E2 for 24 hr.

**Figure supplement 2.** FACS profiles of MCF7 cells treated with 0 nM E2, 1 nM E2, or 100 nM E2 for 24 hr.

**Figure supplement 3.** Effect of E2 on MCF10A cells.

**Figure supplement 4.** Quantification of the percent of cells positive for EdU incorporation after treatment with 0 or 100 nM E2 for the indicated length of time and then pulsed with 10 µM EdU for 30 min prior to fixation.

**Figure supplement 5.** Effects of PHA 767491 on EdU incorporation and Ser2 phosphorylation of RNA Pol II in MCF7 cells.

**Figure supplement 6.** Quantification of the percent of cells positive for EdU incorporation for cells treated with 0 or 100 nM E2 for 12 hr prior to the addition of 0.8 µM flavopiridol or DMSO for 2 hr.

treated with no E2 or 100 nM E2 for 2 or 24 hr. These times were chosen to assess R-loop formation prior to and/or concurrent with E2-transcriptional induction and DNA damage. In mock-treated MCF7 cells, we observed 17,445 DRIP peaks across the genome (*Figure 2—figure supplement 1A*, *Figure 2—figure supplement 1—source data 1*), a number in line with the previous DRIP-seq experiments performed in human Ntera2 cells and primary human fibroblasts (*Ginno et al., 2012*; *Lim et al., 2015*). Strikingly, however, in cells treated with E2 we observed a dramatic increase in DRIP peaks, with 33,458 or 24,781 peaks at 2 or 24 hr, respectively (*Figure 2—figure supplement 1A*, *Figure 2—figure supplement 1—source data 2*, *3*). DRIP peaks from mock-treated cells covered 49.0 Mb of genome space, while those from E2-treated cells for 2 hr or 24 hr covered 81.9 MB and 63.2 Mb of genome space, respectively. Importantly, the vast majority of peaks were abolished by RNase H treatment (*Figure 2—figure supplement 1B*).

Next, we asked whether the RNA-DNA hybrids identified exhibit known R-loop features. Indeed, DRIP peaks were enriched at promoters, 5' gene regions and sites of transcriptional termination (*Figure 2B*), consistent with their published distribution and roles at these regulatory regions (*Ginno et al., 2013*; *Ginno et al., 2012*; *Skourti-Stathaki et al., 2014*). DRIP peaks also exhibited GC skew (*Figure 2—figure supplement 1C*) and overlapped with in vitro mapped G-quartets (*Figure 2—figure supplement 1D*) (*Chambers et al., 2015*), two sequence characteristics that are associated with R-loop formation (*Duquette et al., 2004*; *Ginno et al., 2013*). Taken together, these observations suggest that E2 treatment is associated with a dramatic induction of R-loops at new genomic sites.

To further explore the effects of E2 on R-loop formation, we sought to identify the genomic regions in which DRIP signal was induced at 2 and 24 hr by differential peak calling. Using triplicate data for each condition, we combined all DRIP-positive regions in at least one sample with a set of similarly-sized intervals with no significant enrichment and performed the differential analysis. Differential peaks were only identified if they showed a significant change upon E2 induction in each biological replicate. Analysis of RNA-DNA hybrids induced after E2 treatment revealed a dramatic increase throughout the genome, with 8,023 and 3,263 peaks significantly induced at 2 hr or 24 hr, respectively (*Figure 2C*, *Figure 2—figure supplements 1A,B*, *Figure 2C — source data 1*, *2*). Of those hybrids induced at 24 hr, ~60% were also significantly induced at 2 hr, suggesting that the E2 treatment can result in a sustained increase in R-loop formation at many loci. Loci that showed a robust increase following E2 addition include XBP1, CCND1 (Cyclin D1), and SLC7A5 (*Figure 2D,E*, *Figure 2—figure supplement 1E*), all targets of E2-transcription (*Honkela et al., 2015*). We validated our DRIP-seq data at several loci using DRIP-qPCR. A dose-dependent increase in DRIP signal was observed upon both 10 nM and 100 nM E2 treatment at several E2-transcription targets (*Figure 2F*), while two genomic regions (MLKL and 83/84) devoid of DRIP peaks showed no change.

Using GREAT, a bioinformatic tool that annotates ontologies for cis-regulatory regions of the genome (*McLean et al., 2010*), we next asked whether there was a common signature among these E2-induced RNA-DNA hybrids. We found a dramatic enrichment of R-loops in E2-transcriptionally

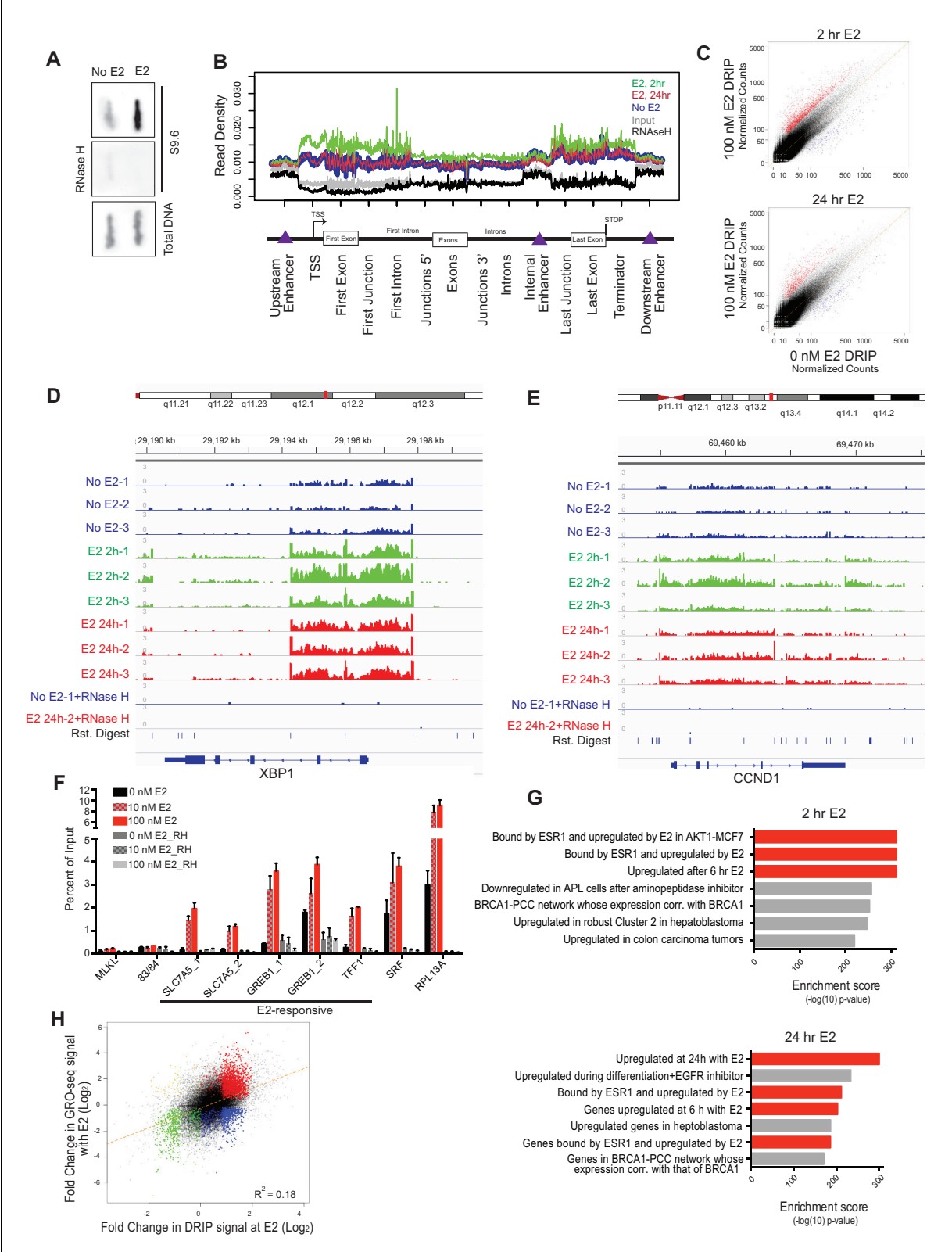

**Figure 2.** Estrogen induces robust R-loop formation at E2-responsive genes. (**A**) Slot blot to detect global RNA-DNA hybrids with S9.6 antibody in MCF7 cells treated with 0 or 100 nM E2 for 24 hr. Total denatured DNA is stained with a single-strand DNA antibody. RNase H was added as indicated. (**B**) Meta-gene analysis for indicated DRIP signal over indicated genomic features. Data are shown for DRIP-seq biological replicate # 3. An enrichment is seen in all data sets but relative read density at these sites varies between replicates. (**C**) DRIP-seq read counts normalized for total mapped reads

*Figure 2 continued on next page*

*Figure 2 continued*

from DRIP in 0 nM E2 conditions vs DRIP from cells treated with 100 nm E2 for 2 hr (top) or 24 hr (bottom). Graphs are from 3 biological experiments. Black dots indicate DRIP peaks and red dots indicate induced DRIP peaks relative to 0 nM E2. (D) Integrated Genome Viewer (IGV) display of DRIP-seq enrichment at XBP1. Scale = million mapped reads. RNase H was performed prior to DRIP-seq on one replicate each from 0 nM E2 and 100 nM E2 24 hr. Independent replicates are shown as 1–3. (E) IGV display of CCND1 (Cyclin D1), as described in (D). (F) DRIP-qPCR validation. Cells were treated with 0, 10, or 100 nM E2 for 24 hr and harvested for DRIP. MLKL and 83/84 are negative controls. Error bars represent S.E.M. of 2 biological experiments. RNase H treatment was performed for 24 hr prior to DRIP-qPCR where indicated. (G) Functional signatures by GREAT of E2-induced DRIP peaks found to be differentially induced in 100 nM E2, 2 hr (top) or 100 nM E2 24 hr (bottom) than in 0 nM E2 treated cells. The 7 highest enrichment scores are shown, with red highlighting E2-associated signatures. (H) Fold change in DRIP signal after 2 hr of 100 nM E2 relative to 0 nM E2 (x-axis) vs. fold change in GRO-seq signal after 160 min of 100 nM E2 relative to 0 nM E2 (y-axis). E2-induced DRIP peaks that show a positive (red) or negative (blue) fold change in GRO-seq upon E2 stimulation are highlighted. Negative changes in DRIP upon E2 that correspond to a positive (yellow) or negative (green) fold change in GRO-seq are also shown. GRO-seq data from (*Hah et al., 2011*).

The following source data and figure supplements are available for figure 2:

**Source data 1.** Genomic coordinates for DRIP peaks identified as induced in MCF7 cells treated with 100 nM E2 for 2 hrs relative to MCF7 cells treated with 0 nM E2.

**Source data 2.** Genomic coordinates for DRIP peaks identified as induced in MCF7 cells treated with 100 nM E2 for 24 hrs relative to MCF7 cells treated with 0 nM E2.

**Figure supplement 1.** R-loops are induced with E2 prior to S phase and exhibit R-loop features.

**Figure supplement 1—source data 1.** Genomic coordinates for all identified DRIP peaks from MCF7 cells treated with 0 nM E2.

**Figure supplement 1—source data 2.** Genomic coordinates for all identified DRIP peaks from MCF7 cells treated with 100 nM E2 for 2 hrs.

**Figure supplement 1—source data 3.** Genomic coordinates for all identified DRIP peaks from MCF7 cells treated with 100 nM E2 for 24 hr.

**Figure supplement 2.** Sequence features and expression analysis associated with DRIP-seq.

responsive genes at both time points (*Figure 2G*, red highlighting). These data clearly demonstrate that the induction of R-loops by E2 is a robust response that occurs quickly at E2-responsive genes, and persists at a subset of these genes. Taken together with data shown in *Figure 1*, these data also demonstrate that R-loop formation occurs prior to the onset of DNA damage, and in cells that have not yet entered S phase (*Figure 2—figure supplement 1F*).

To systematically investigate the relationship between E2-responsiveness and R-loop induction, we also correlated DRIP signal enrichment with changes in expression level upon E2 stimulation. We focused on transcriptionally engaged RNA polymerase by using publically available global run-on sequencing (GRO-seq) data from samples prepared under similar conditions (*Hah et al., 2011*) (*Figure 2H*). A modest correlation between the induction of DRIP signal and the induction of expression was observed, consistent with the requirement for transcription in R-loop formation. Although the majority of induced DRIP peaks were found in E2-responsive genes (red highlighting), some were also induced at sites that were not E2-responsive or that were negatively regulated by E2 (blue highlighting). Both sets of hybrid-forming sequences exhibit GC skew and enhanced G-quadruplex formation (*Figure 2—figure supplements 2A,B*). We also found no meaningful correlation between DRIP signal enrichment and overall expression level as measured by publically available RNA-seq data from similarly paired conditions. This finding indicates that DRIP peaks do not form only on highly expressed genes (*Figure 2—figure supplement 2C*). We conclude that R-loop induction by E2 is correlated with E2-induced transcription, but that expression level is not the sole determinant of R-loop formation and other secondary structures or sequence elements may contribute.

## Breast cancer rearrangements are enriched in E2-responsive genes

Because E2-induced R-loops are strongly enriched in E2-transcriptionally responsive genes, we wondered if estrogen-responsive loci are also more prone to mutation in breast cancer. To accomplish this, we utilized the somatic mutation data from the recent, large-scale whole-genome sequencing

of 560 breast tumors (*Nik-Zainal et al., 2016*). Three groups of somatic mutations were used: 1) structural variants such as duplications, deletions, or inversions, 2) translocations, or 3) simple somatic mutations such as substitutions and indels (*Nik-Zainal et al., 2016*). For each group of mutations, we determined whether the sites of mutation were enriched in E2-responsive loci, as measured by the Z-score from RNA-seq data (*Honkela et al., 2015*). For mutated genes in each of the three classifications, we examined enrichment relative to control gene sets matched for both expression level and replication timing (*Figure 3—figure supplements 1*, *2,* and *3*). Notably, we observed a significant enrichment of both structural variants ($p<1.00*10^{-6}$) and translocations ($p<1.00*10^{-6}$) in genes that are transcriptionally-responsive to E2 (*Figure 3A,B*). In contrast, there were fewer simple somatic mutations in E2-responsive loci relative to the matched set ($p<1.00*10^{-6}$) (*Figure 3C*). These results demonstrate that genes whose expression is induced in response to E2 are enriched in genomic rearrangements in breast tumors and support the view that these sites are prone to certain types of DNA damage.

## E2-induced R-loops colocalize with DNA damage markers on chromatin

Given that E2-responsive genes are strongly enriched in regions of genomic rearrangement in breast tumors and these mutations could arise through DNA DSB formation, we asked whether we could directly detect DNA damage in the vicinity of E2-induced R-loops. To address this, we performed a proximity ligation assay (PLA) using the S9.6 and P-H2AX antibodies to visualize interactions between RNA-DNA hybrids and phosphorylated H2AX. Notably, the percentage of cells that contain at least one PLA focus in the nucleus increased approximately 2-fold following E2 treatment (*Figure 4A,B*). The relatively small number of detectable interactions between the RNA-DNA hybrid and P-H2AX antibodies upon E2 addition may indicate that the majority of DNA damage occurs at a distance from the R-loop that is outside the limits of PLA detection (<30–40 nM). In order to confirm the interaction between R-loops and P-H2AX, we used the S9.6 antibody to precipitate RNA-DNA hybrids from cross-linked E2-treated cells. We found that that both P-H2AX and another DDR target, P-KAP1, interacted with RNA-DNA hybrid containing chromatin fragments (*Figure 4C*). This interaction was specific, as it did not occur with MCM3, another chromatin-bound protein (*Figure 4C*). Taken together, these results suggest that DNA damage markers bind the chromatin environment near R-loops following E2 treatment.

## RNase H reduces E2-induced DNA damage and DSBs

We then asked whether the E2-induced accumulation of DSBs is a direct result of E2-induced R-loop formation. If R-loops cause E2-induced DNA damage, expression of RNase H should abrogate DSB formation. We tested this idea by monitoring P-H2AX levels in an MCF7 cell line stably expressing a doxycycline-inducible, Flag-tagged RNase H (*Figure 4D*). Strikingly, we observed a dramatic and dose-dependent reduction in P-H2AX intensity and foci formation upon increasing RNase H expression (*Figure 4E,F*, *Figure 4—figure supplement 1*). Similar results were observed in two additional independent stable cell clones (*Figure 4—figure supplements 2A,B*). Importantly, expression of RNase H did not alter E2-driven cell cycle progression (*Figure 4—figure supplement 3*), nor did it reduce global levels of transcription, as measured by EU labeling (*Figure 4—figure supplements 4A,B*). Thus, the decreased P-H2AX signal is not a result of reduced transcription or a defect in cell cycle progression. Finally, we found that RNase H expression resulted in a significant decrease in DSBs upon E2 stimulation, as measured by the neutral comet assay (*Figure 4G,H*). These findings clearly demonstrate that RNA-DNA hybrids are needed to form E2-induced DSBs.

## NER factors are required for E2-induced DSB formation

We recently showed that R-loop-mediated DSB formation depends on factors specifically involved in transcription-coupled nucleotide excision repair (TC-NER), but not factors specifically involved in global genome NER (GG-NER) (*Sollier et al., 2014*). In these studies, R-loops were induced by the knockdown of RNA processing genes and/or splicing factors. To ask if the DNA damage resulting from E2-induced co-transcriptional R-loop formation arises by a similar mechanism, we assessed whether knockdown of NER factors altered the DDR following E2. Strikingly, we found that P-H2AX levels were significantly reduced following knockdown of XPG, an endonuclease involved in both forms NER (*Figure 5A*, *Figure 5—figure supplement 1*). XPG knockdown also dramatically reduced

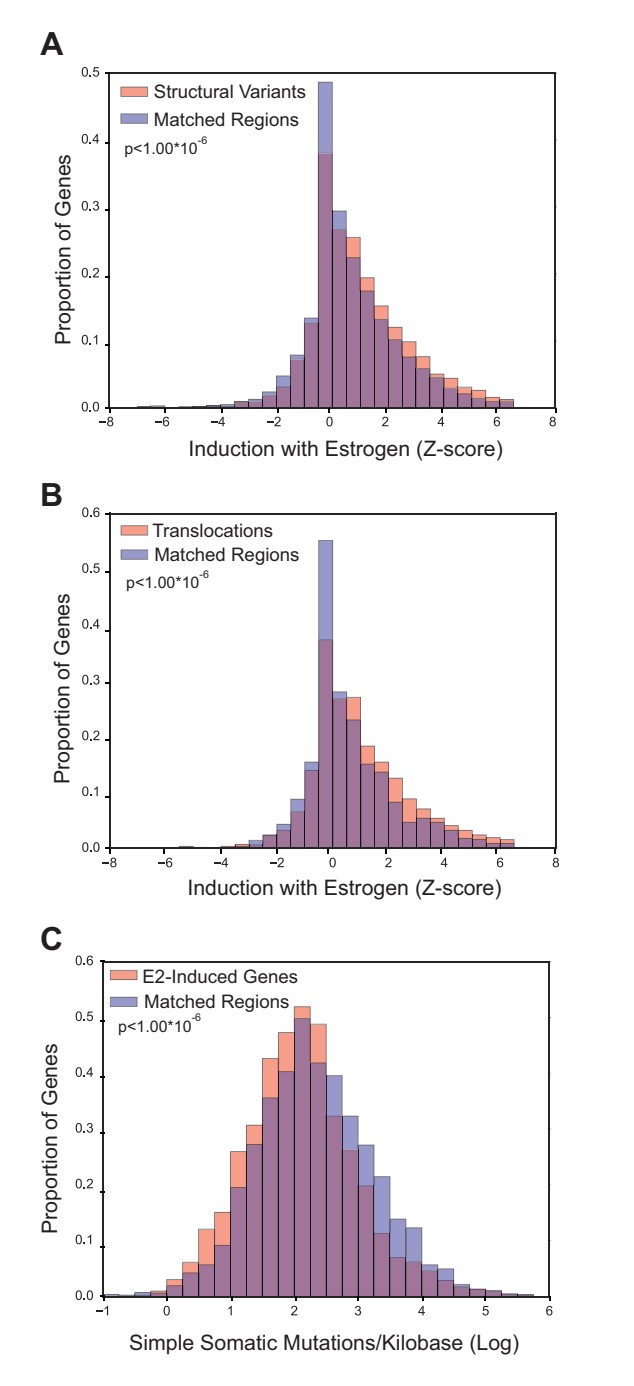

**Figure 3.** Breast cancer rearrangements are enriched in E2-responsive genes. (**A**) Histogram of E2-induced expression changes (Z-score) for genes that overlap with breast cancer structural variants (red bars) compared to the distribution of E2-induced expression changes (Z-score) for a control gene set (blue bars), matched for expression level and replicating timing. Genes with structural variants are enriched in E2-responsive genes. $p < 1.00 \times 10^{-6}$. (**B**) Histogram of E2-induced expression changes (Z-score) for genes with breast cancer translocations (red bars) compared to a control gene set (blue bars), as described in (**A**). Genes with translocations are enriched in E2-responsive genes $p < 1.00 \times 10^{-6}$. (**C**) Histogram of enrichment of breast cancer simple somatic mutations in E2-responsive genes. Data plotted as the log transformation of simple somatic mutations per kilobase for E2-induced genes (red bars) relative to a matched set (blue bars), as described previously. E2-responsive genes have fewer simple somatic mutations than the control gene set ($p < 1.00 \times 10^{-6}$). In **A–C**, p-values represent two-tailed bootstrap of medians. All breast cancer mutation data from (**Nik-Zainal et al., 2016**).
*Figure 3 continued on next page*

*Figure 3 continued*

The following figure supplements are available for figure 3:

**Figure supplement 1.** Comparison of (**A**) Replication timing based on RepliSeq signal across the gene body of regions containing structural variants in breast cancer (x-axis) relative to that of the matched regions (y-axis), and (**B**) Mean expression level based on RNA-seq of regions associated with structural variants (x-axis) relative to that of the matched regions (y-axis).

**Figure supplement 2.** Comparison of (**A**) Replication timing based on RepliSeq signal across the gene body of translocated regions (x-axis) relative to that of the matched regions (y-axis), and (**B**) Mean expression level based on RNA-seq of translocations (x-axis) relative to that of the matched regions (y-axis).

**Figure supplement 3.** Comparison of (**A**) Replication timing based on RepliSeq signal across the gene body of genes with simple somatic mutations (x-axis) relative to that of the matched regions (y-axis), and (**B**) Mean expression level based on RNA-seq of genes with simple somatic mutations (x-axis) relative to that of the matched regions (y-axis).

---

the comet tail moment that forms in response to E2 treatment (*Figure 5B,C*), signifying a significant reduction in E2-mediated DSBs. Importantly, we also found that knockdown of the TC-NER specific factor Cockayne syndrome group B (CSB) significantly reduced P-H2AX levels induced by E2 (*Figure 5D*). In contrast, knockdown of XPC, a lesion-recognition factor specifically involved in GG-NER, caused little change in P-H2AX levels after E2 treatment (*Figure 5E*). These findings suggest that R-loops induced by E2 are processed by TC-NER factors. Taken as a whole, our findings demonstrate that replication-dependent E2-induced DNA damage results from the induction of R-loops triggered by E2-mediated transcription.

## Discussion

Given the clear link between E2 exposure and breast cancer incidence, there is significant interest in understanding the molecular mechanisms by which E2 promotes DNA damage and genome instability, which can drive tumorigenesis. Accordingly, several mechanisms for E2-induced DNA damage have been elucidated. For example, TopoIIβ and APOBEC3B cause the rapid and transient formation of DNA damage in a process necessary for E2-induced transcriptional activation (*Ju et al., 2006*; *Periyasamy et al., 2015*). Additionally, E2 metabolites can cause oxidative DNA damage (*Lavigne et al., 2001*; *Yager, 2014*). Here, we report the first R-loop-dependent mechanism of E2-induced DNA damage. Specifically, we show that R-loops induced by E2 treatment in ER-positive breast cancer cells lead to DSB formation in the vicinity of the R-loop. Our findings suggest that increased transcription due to a physiologically relevant stimulus may result in R-loop-mediated genomic instability.

Intriguingly, the R-loop-dependent DNA damage we observe requires the initiation of DNA replication (*Figure 1F,H*). Although E2 causes a dramatic increase in R-loops after 2 hr (*Figure 2C*), no significant change in DNA damage is observed at that time (*Figure 1F*). Thus, although R-loops are present, they do not cause DNA damage until S phase entry. Consistent with this finding, R-loop-dependent DNA damage has been previously associated with DNA replication (*Alzu et al., 2012*; *Gan et al., 2011*; *Houlard et al., 2011*). Moreover, a few common fragile sites correspond to late-replicating, long genes where R-loops form, and early-replicating fragile sites are associated with highly transcribed genes (*Barlow et al., 2013*; *Helmrich et al., 2011*). This suggests that genes that are highly transcribed or R-loop prone may pose the greatest challenge to genome integrity when they meet the replication fork. Interestingly, we find that TC-NER factors are also involved in generating the replication-associated DNA damage observed (*Figure 5*). This raises the possibility that these factors may be acting in a non-canonical manner in S phase, and there are several possible mechanisms by which their activity may be coordinated with replication activities that have been previously discussed (see *Sollier et al., 2014*).

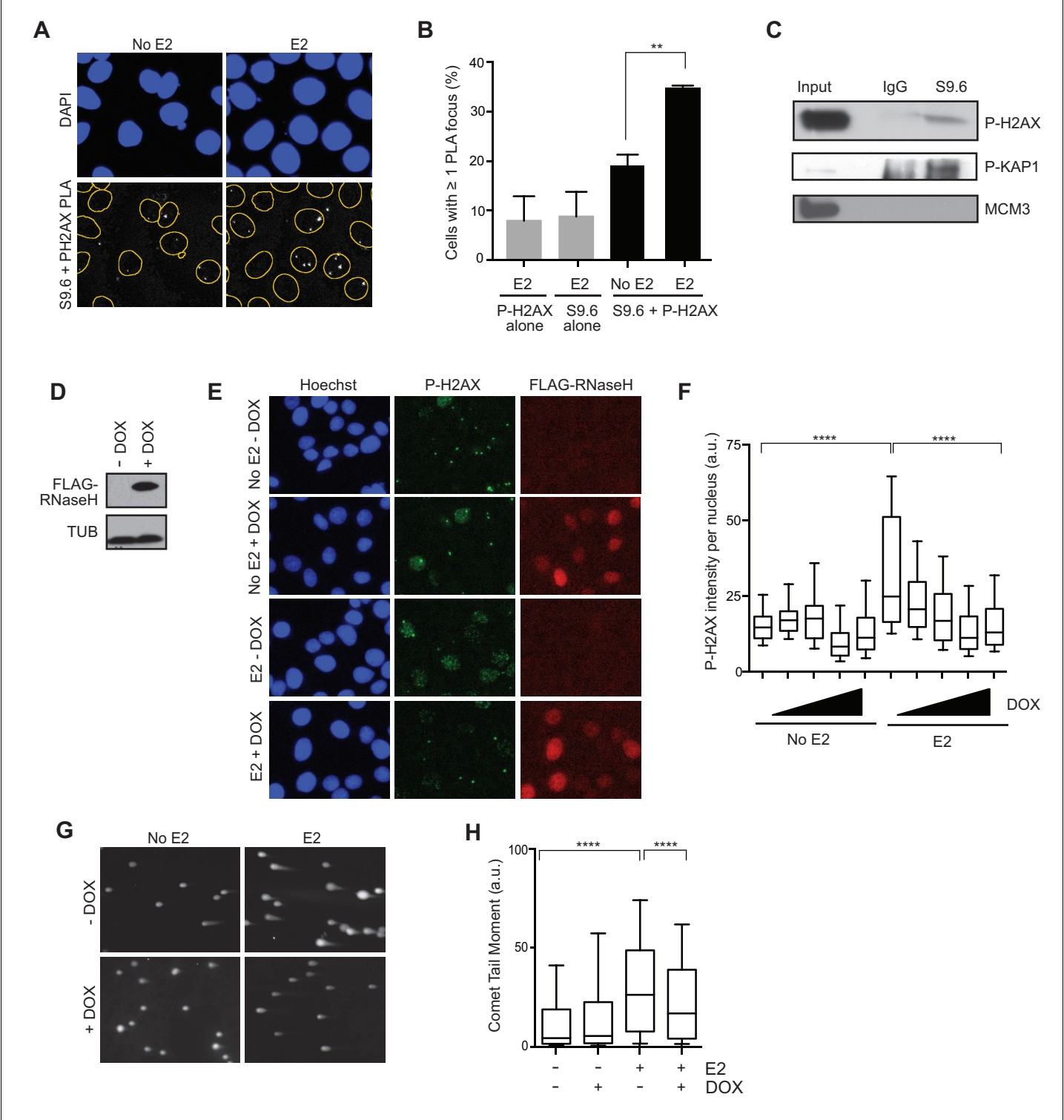

**Figure 4.** E2-induced R-loops occur on chromatin marked by DNA damage and RNase H reduces E2-induced DSBs. (A) Proximity ligation assay between S9.6 antibody and P-H2AX antibody in cells treated either with 0 or 100 nM E2 for 24 hr. (B) Quantification of the percentage of cells with ≥ 1 PLA focus per nucleus. Single-antibody controls from cells treated with 100 nM E2 for 24 hr are shown. Error bars represent the SEM from 4 biological replicates. **p<0.01 (Student's t-test). (C) P-H2AX and P-KAP1 levels from co-IP of S9.6 or IgG from cross-linked and sonicated cells treated with 100 nM E2 for 24 hr. Input is 60% of the IP. (D) Western blot with Flag antibody in MCF7 tetOn-RH cells. 1000 ng/mL doxycycline (DOX) was added for 48 hrs where indicated. (E) Immunostaining for P-H2AX and FLAG in MCF7 tetON-RH cells treated with increasing concentrations of DOX (100, 250, 500 1000 ng/mL) for 24 hrs prior to the addition of either 0 or 100 nM E2 for 24 hr. (F) Quantification of P-H2AX intensity for the experiment described in (E),

*Figure 4 continued on next page*

*Figure 4 continued*

where the triangle indicates increasing concentrations of DOX. ****p<0.0001 (non-parametric Mann-Whitney rank sum t-test). n = 3 biological replicates. >1000 cells/condition quantified. (G) Neutral comet assay in MCF7-tetOn-RH cells treated with or without 1000 ng/mL DOX for 24 hrs prior to 0 nM E2 or 100 nM for 24 hr. (H) Quantification of the neutral comet tail moment described in (G). **p<0.01 (non-parametric Mann-Whitney rank sum t-test). n = 3 biological replicates. >100 comets/condition. a.u. = arbitrary units.

The following figure supplements are available for figure 4:

**Figure supplement 1.** Quantification of the fold change in the percent of cells with >5 P-H2AX foci per cell in MCF7-tetON-RH cells treated with indicated concentrations of DOX. *p<0.05.

**Figure supplement 2.** Expression of RNase H prevents E2-induced DNA damage.

**Figure supplement 3.** FACS profiles of MCF7 tetON-RH cells treated with or without 1000 ng/mL DOX for 24 hr prior to the addition of 100 nM E2 for 24 hr.

**Figure supplement 4.** EU incorporation following RNase H expression in MCF7 cells.

Importantly, we find that R-loops are induced predominantly at E2-responsive genes ($p<10^{-233}$). In agreement with this, we see a correlation between the induction of expression by E2, as measured by GRO-seq, and the induction of R-loops by E2 (*Figure 2H*). The modest strength of the correlation likely indicates that factors distinct from the transcription status play an important role in dictating R-loop formation. Interestingly, a small subset of genomic regions exist that show a significant increase in R-loops despite no increase in expression (*Figure 2H*, blue highlighting). Similarly, RPL13A, the expression of which is unaffected by E2 (*Shah and Faridi, 2011*), showed a significant increase in R-loops upon E2 (*Figure 2F*). One possibility is that upon a burst in transcription, the cell's normal splicing and R-loop modulating factors initially become saturated. Thus, some loci that are prone to form R-loops may accumulate these structures and become vulnerable to DNA damage, regardless of their response to the transcription stimulus.

We also observe that breast cancer rearrangements, but not simple somatic mutations, are significantly enriched in E2-responsive genes (*Figure 3*). Both structural variants and translocations have been suggested to occur due to erroneous repair of a DSB (*Kasparek and Humphrey, 2011*) and these mutation types have been shown to be more prevalent in breast tumors relative to other tumor tissues (*Yang et al., 2013*). Intriguingly, some cancer-associated genes are found among genes forming E2-induced R-loops. For example, we observed a dramatic increase in R-loops at Cyclin D1 (*Figure 2E*), which is amplified in up to 20% of breast cancers, the majority of which are ER-positive (*Arnold and Papanikolaou, 2005*; *Osborne et al., 2004*). Similarly, E2 induces robust R-loop formation at ZNF703. This locus has been implicated in driving oncogenesis in luminal B breast tumors (*Holland et al., 2011*; *Sircoulomb et al., 2011*). Both Cyclin D1 and ZNF703 were among the genes identified as breast cancer drivers, as were an additional 27 genes with E2-induced R-loops (*Nik-Zainal et al., 2016*). Our findings raise the possibility that E2-induced R-loop formation could lead to DNA damage and genome instability at some of these and other R-loop prone loci. Intriguingly, testosterone (DHT) stimulation in prostate cells has been shown to lead to DSBs and recurrent translocations at androgen-receptor (AR) target loci (*Haffner et al., 2010*; *Lin et al., 2009*). While these breaks are TOP2B-dependent and necessary for the induction of transcription, DHT may also induce DNA damage in an R-loop and replication dependent manner. Notably, in prostate cancer, recurrent rearrangements are frequently observed at androgen-receptor (AR) target genes (*Kumar-Sinha et al., 2008*).

In summary, our findings have uncovered a role for R-loops in E2-induced DNA damage and reveal a new mechanism by which genome instability may arise in ER-responsive breast cancer cells. Notably, transcription factor responses vary between cell types and following different physiological and environment stimuli. Thus, the induction of potentially toxic R-loops at sites of induced transcription may offer insight into the tissue-specific DNA damage and mutation patterns observed in some cancers.

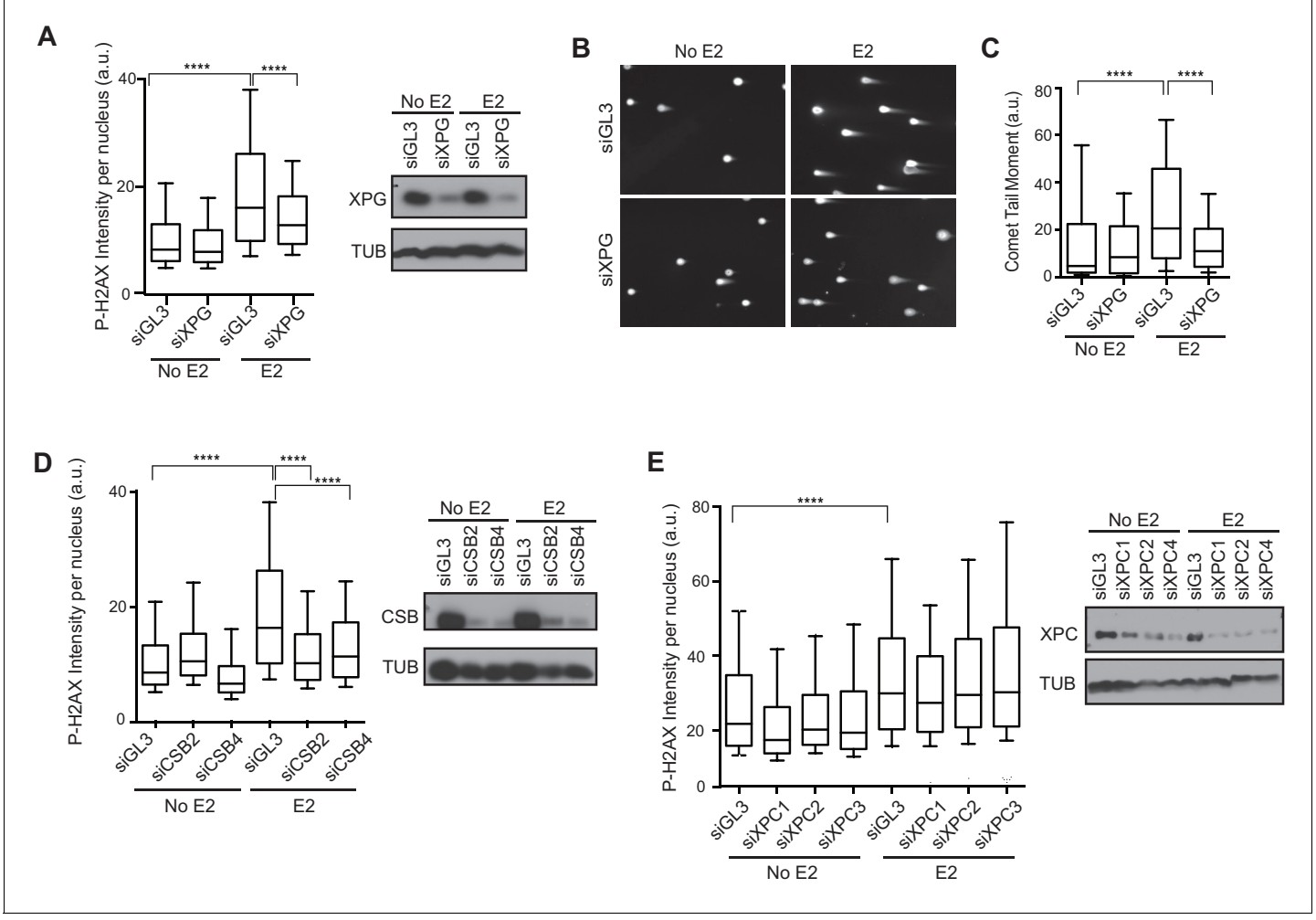

**Figure 5.** Knockdown of NER and R-loop processing factors reduces E2-induced DNA damage and DSBs. (A) P-H2AX intensity based on immunostaining of MCF7 cells transfected with indicated siRNA 64 hrprior to the addition of 0 or 100 nM E2 for 24 hr. ****p<0.0001. Western blot shows the level of XPG. (B) Neutral comet assay in cells transfected with indicated siRNA 64 hr prior to the addition of 0 or 100 nM E2 for 24 hr. (C) Quantification of neutral comet tail moment described in (B). ****p<0.0001. >100 comets/condition. (D, E) P-H2AX intensity based on immunostaining in MCF7 cells treated as in (A) with the indicated siRNA. ****p<0.0001. Western blot shows the level of CSB or XPC. The siGL3 control in the quantification shown in (D) is the same as shown in (A). a.u. = arbitrary units. For all data, associated p-values are from non-parametric Mann-Whitney rank sum t-test. n=3 biological replicates. Quantification from >1000 cells/condition unless noted.

The following figure supplement is available for figure 5:

**Figure supplement 1.** XPG knockdown does not alter E2-induced cell cycle progression.

# Materials and methods

## Cell culture

MCF7 cells were obtained from ATCC (HTB-22, RRID:CVCL_0031), where they were authenticated by STR profiling and tested negative for mycoplasma. These MCF7 cells were cultured in DMEM (GIBCO) supplemented with 10% FBS in 5% $CO_2$ at 37°C. For estrogen stimulation, MCF7 cells were plated in DMEM with 10% FBS for at minimum of 16 hr before being washed 3 times in 1X PBS. Phenol-red free DMEM (GIBCO) with 10% charcoal-stripped FBS (Invitrogen) was then added for 48 hr prior to the addition of E2 dissolved in EtOH or the vehicle control (100% EtOH). MCF10A cells were obtained from ATCC (CRL-10317, RRID:CVCL_0598), where they were authenticated by STR profiling and tested negative for mycoplasma. MCF10A cells were cultured in a 1:1 mixture of

DMEM and F12 medium (GIBCO) supplemented with 5% horse serum, hydrocortisone (0.5 mg/ml), insulin (10 µg/ml), epidermal growth factor (20 ng/ml), cholera toxin (100 ng/ml) and penicillin-streptomycin (100 µg/ml each). For estrogen stimulation, MCF10A cells were washed 3 times in 1X PBS and then media containing a 1:1 mixture of phenol-red free DMEM and F12 medium (GIBCO) supplemented with 5% charcoal-stripped FBS (Invitrogen), hydrocortisone (0.5 mg/ml), insulin (10 µg/ml), and cholera toxin (100 ng/ml) was added and cells were grown for 48 hr before stimulation. The MCF7-TetON-RH cell line was generated from MCF7 cells (ATCC HTB-22, RRID:CVCL_0031, as described above) that were first transduced with a pLVX-EF1α-Tet3G-Hygro transactivator-containing vector and selected with 500 µg/mL hygromycin. Hygromyocin-resistant cells were then transduced with a pLVX-Tight-Puro vector, expressing a FLAG-tagged truncated version of RNase H1 (pLVX-Tight-Puro-RH-Flag) and selected with 1 µg/mL puromycin. Cells were maintained in 500 µg/mL hygromyocin and 1 µg/mL puromycin.

## Antibodies, plasmids, and reagents

Antibodies to single-stranded DNA (EMD Millipore, #MAB3868), P-H2AX (Cell Signaling, 9718S), P-KAP1 (Bethyl, A300-767A), MCM3 (Abcam, ab4460), XPG (Santa Cruz, 393004), CSB (Bethyl, A301-345A), XPC (Abcam, ab6264), 53BP1 (BD, 612523), BrdU (BD, 347580) ALPHA-TUB (Sigma, T9026), FLAG (Sigma, F1804), GAPDH (Abcam, ab8245), ER-α (Santa Cruz, HC-20), and RNA Pol II phospho S2 H5 (Abcam, ab24758) were used. The S9.6 antibody was purified from a S9.6 hybridoma cell line (ATCC, HB-8730; RRID: CVCL_G144). Hybridoma supernatant was applied to a 1 mL HiTrap Protein G HP column (GE Healthcare). The antibody was eluted with 100 mM glycine pH 2.5, in 0.5 mL fractions. Fractions were screened for antibody by SDS-PAGE and antibody-containing fractions were pooled and dialyzed in PBS overnight followed by dialysis in 50% glycerol for 6 hr. The antibody concentration was measured against a BSA standard and aliquots were made at 1 µg/mL. The pLVX-Tight-Puro-RH-Flag was made by inserting a truncated FLAG-tagged human RNase H1 lacking the first 27 amino acids. siRNAs, purchased from ThermoFisher, were: siGL3 (D-001400-01), siXPG (D-006626-02), siCSB2 (D-00488-04), siCSB4 (D-00488-06), siXPC1 (D-016040-01), siXPC2 (D-016040-02), and siXPC3 (D-016040-04). All siRNA transfections were performed using Dharmafect1 (ThermoFisher) according to the manufacturer's protocol and 20 nM siRNA unless otherwise indicated. For siRNA transfections, cells were reverse transfected with 20 nM siRNA in antibiotic free DMEM with 10% FBS. 16 hr later, the media was removed, cells were washed 3 times with 1X PBS and phenol-red free DMEM with 10% charcoal-stripped FBS was added. 64 hr after transfection, cells were transfected again with 10 nM of the siRNA and E2 or vehicle control (100% EtOH) was added to the media for 24 hrs. 17ß-Estradiol (E2) (Sigma, AC00006) was dissolved in 200-proof EtOH (Goldshield). Flavopiridol (Tocris, 3094) and the Cdc7 inhibitor PHA 767491 (Sigma, PZ0178) were prepared in DMSO. Doxycycline (DOX) (Sigma) was added every 24 hr when indicated. Human RNaseH purified from MBP-RNaseH1 was a gift from the Chedin lab and was used with commercially available RNase H buffer (NEB, M0297).

## Neutral comet assay

The neutral comet assay was performed using the CometAssay Reagent Kit for Single Cell Gel Electrophoresis Assay (Trevigen) according to the manufacturer's protocol with 1X TBE running buffer. Electrophoresis was performed at 4°C in the CometAssay Electrophoresis System II. SYBR-Gold (Invitrogen) was used to stain DNA. Images were taken on an epifluorescence microscope at 10x magnification. Comet tail moments were quantified using OpenComet software (Gyori et al., 2014). In box and whisker plots, box and whiskers indicate 25–75 and 10–90 percentiles, respectively, with lines representing median values.

## Immunostaining

Cells were fixed with 4% PFA/PBS (EMS) for 15 min, permeabilized with 0.25% Triton-X 100 for 15 min, washed 3 times in 1X PBS, and blocked in 2% BSA/PBS for 1 hr at RT. The primary antibody was incubated overnight at 4°C. Antibodies were used at the following dilutions: Rabbit P-H2AX antibody (1:500, Cell Signaling), FLAG, mouse FLAG antibody (1:500, Sigma), mouse 53BP1 (1:500, BD). Cells were then washed 3 times in 1X PBS and stained with Hoechst (1:1000) and anti-rabbit AlexaFluoro-488-conjugated secondary (1:1000). For co-staining, anti-mouse AlexaFluoro-

594-conjugated secondary (1:1000) was used. Cells were imaged on a fully automated Molecular Devices ImageXpress Micro microscope. Images were captured at either 20X or 40X. Analysis of P-H2AX intensity was performed MetaXpress quantification software, where Hoechst is used as a mask for the nucleus and the intensity and/or number of foci per nucleus is determined. In box and whisker plots, box and whiskers indicate 25–75 and 10–90 percentiles, respectively, with lines representing median values.

## EdU staining

For EdU staining, cells were pulsed for 30 min with 10 µM EdU from the Click-iT Edu Alexa Fluor 488 imaging kit (ThermoFisher). Cells were then fixed by 4% PFA/PBS for 15 min, and permeabilized with 0.25% Triton/PBS for 15 min. The Click-It reaction was then performed according to manufacturer's instructions. Immunostaining was carried out as described. Images were acquired on a fully automated Molecular Devices ImageXpress Micro microscope and intensity analysis was performed using MetaXpress quantification software.

## EU staining

For EU staining, cells were pulsed for 1 hr with 100 µM EU from the Click-iT RNA Alexa Fluor 488 imaging kit (ThermoFisher). When indicated, 100 µM DRB (Cayman Chemical Company) was added for 2 hrs with 100 µM EU being added for the last hour. Cells were then fixed by 4% PFA/PBS for 15 min, and permeabilized with 0.25% Triton/PBS for 15 min. The Click-It reaction was then performed according to manufacturer's instructions. Cells were then incubated in Hoechst (1:1000) for 15 min before the slides were mounted with Pro-Long Gold Antifade reagent and imaged on a Zeiss Axioscope at 40X. EU signal intensity was calculated using ImageJ (v 1.47).

## S9.6 Slot blotting

Total nucleic acid was extracted by a standard SDS/Proteinase K lysis followed by phenol/chloroform extraction and EtOH/sodium acetate precipitation. DNA (1 µg) from each sample was spotted on Nylon membrane (Amersham) using a slot blot apparatus and vacuum suction. For RNase H treatment, 5 µg of DNA was treated with RNase H at 37°C overnight, and then purified by standard phenol/chloroform EtOH precipitation. For total DNA control, the membrane was denatured for 10 min in 0.5 M NaOH, 1.5 M NaCl, and neutralized for another 10 min in 1 M NaCl, 0.5 M Tris-HCl pH7.0. Membranes were then UV-crosslinked ($0.12J/m^2$), blocked in 5% milk/TBST, and incubated overnight at 4°C with mouse S9.6 (1:500) or single-strand DNA antibody (1:10,000). Blots were washed 3 times with TBST and secondary antibody (1:10,000 goat anti-mouse HRP) was added for 1 hr at RT.

## S9.6 Immunoprecipitation (DRIP)

DRIP was performed as described in *Ginno et al. (2012)*. Briefly, DNA was extracted carefully by phenol/chloroform extraction in phase lock tubes, precipitated with EtOH/sodium acetate, washed with 70% EtOH, and resuspended in TE. DNA was digested with a cocktail of restriction enzymes (Bsrg1, EcoR1, HindIII, SspI, XbaI) overnight at 37°C. For RNase H-treated samples, 8 µg of DNA was treated with RNase H overnight at 37°C. DNA was purified by standard methods described above. 4 µg of DNA was bound with 10 µg of S9.6 antibody in 1 X binding buffer (10 mM NaPO4 pH 7, 140 mM NaCl, 0.05% Triton X-100) overnight at 4°C. Protein A/G sepharose beads were added for 2 hr. Bound beads were washed 3 times in binding buffer and elution was performed in elution buffer (50 mM Tris pH 8, 10 mM EDTA, 0.5% SDS, Proteinase K) for 45 min at 55°C. DNA was purified as described. qPCR was performed on a Roche LightCycler 480 Instrument II using SYBR-Green master mix (Biorad). qPCR primers are described in *Supplementary file 1*.

## Library preparation and sequencing

DNA libraries were prepared from 3 pooled DRIPs by sample. Briefly, input and DRIP DNA was sonicated to approximately 300–700 bp on a Bioruptor (Diagenode). End repair (NEB E6050S), A-tailing (NEB M0212S), ligation of adapters (NEB M2200S; Affymetrix Prep2Seq Adapters 79800), and PCR amplification were performed by standard methods as described (*Ginno et al., 2012*). Ampure XP beads (Beckman Coulter, A63880) were used for size selection. Libraries were pooled and

sequenced on an Illumina HiSeq machine with single-end 50 bp reads by Elim BioPharm (Hayward, CA).

## Peak calling and differential peak analysis

Raw reads from the DRIP experiments were aligned to the reference genome hg19/GRCh37 using bowtie2. An interval file of restriction fragments for the enzyme cocktail used in DRIP was obtained by verbose search of restriction site sequences with bowtie. Aligned reads with Q score over 10 were counted over the intervals separated by restriction enzyme cut sites using bedtools. Intervals with lower coverage (less than 10 counts per interval in any sample) were removed from the data set. Regions enriched over background were discovered with DESeq2. To perform differential analysis using triplicate data, we combined the set of regions with positive DRIP signal in at least one condition, together with the matched set of negative intervals and performed differential analysis with DESeq2. Read counts were normalized to the total number of mapped reads. To perform analysis of functional term enrichment, we ran GREAT analysis (*McLean et al., 2010*) on sets of differential DRIP regions.

## Expression correlation analysis

RNA-seq data from GEO, accession number GSE62789, was obtained from MCF7 cells that were hormone starved and treated with 0 nM E2 or 10 nM E2 for 160 min (used to match 2 hr DRIP) or 1280 min (used to match 24 hr DRIP) (*Honkela et al., 2015*). DRIP peaks were sorted for those that are within 1 KB of a gene. 7628 gene-proximal regions that form DRIP peaks in any sample were identified. XY plots were generated in R for both control and E2 conditions showing the relative expression distribution of all genes relative to $Log_2$Fold change in DRIP peak signal over input.

## GRO-seq analysis

GRO-seq data from MCF7 cells was obtained from (*Hah et al., 2011*), where cells were treated with vehicle or 100 nM E2 for 160 min. The bedGraph data from accession GSE27463 was converted to coverage files over the DRIP restriction intervals using bedtools. As GSE27463 is aligned to hg18, liftOver was used to convert the intervals to hg18 before determining the coverage. Fold changes between the mock and 160 min E2 GRO-seq were determined using DESeq2. Intervals that had an adjusted p value < 0.1 in the DRIP and an adjusted p value < 0.5 in the GRO-seq were called as significant, and were split into categories on the basis of having positive or negative log2 fold changes in DESeq2. The criteria used when in *Figure 2H* are: (1) Up in DRIP: log2FoldChange(2 hr IP vs Mock IP) > 0, adjusted p<0.1 (2) Down in DRIP: log2FoldChange(2 hr IP vs Mock IP) < 0, adjusted p<0.1 (3) Up in GRO-seq: log2Fold Change(2 hr GRO-seq vs Mock GRO-seq) > 0, adjusted p<0.9 (4) Down in GRO-seq: log2Fold Change(2 hr GRO-seq vs Mock GRO-seq) < 0, adjusted p<0.9.

## Cell cycle analysis

Cells were pulse-labeled with 25 µM BrdU for 30 min. Cells were washed with 1X PBS, and the cell pellet was resuspended in 500 µL 1X PBS. Cells were fixed in 5 mL ice-cold 70% EtOH, permeabilized with 0.25% TritonX-100/PBS for 15 min on ice, blocked in 2% BSA/PBS for 15 min, and incubated in primary BrdU antibody (BD Bioscience 1:100) for 3 hr. Cells were then washed with 1X PBS 3 times and incubated for 1 hr in a 1:400 solution of AlexaFluoro-488 secondary. Propidium iodide (PI, 0.1 mg/mL, Sigma) and RNase A (Qiagen) were added and cells were run on a FACS Caliber (BD Biocience. Cell cycle profiles were determined using FlowJo software.

## Breast cancer mutation analysis for enrichment in E2-responsive genes

For all genes in the MCF7 mock and E2-induced RNA-seq dataset (*Honkela et al., 2015*), E2-induction was defined as the z-score between the mock-treated (0 hr) and E2-induced samples following 160 min E2. The z-score was calculated using the provided means and standard deviations from the dataset, using the formula $(\mu_{E2} - \mu_{Mock})/sqrt(\sigma^2_{E2} + \sigma^2_{Mock})$. Breast cancer structural variants, translocations, and simple somatic mutations were obtained from (*Nik-Zainal et al., 2016*), accession number EGAS00001001178.

For the translocation data, genes were included on the list if they had at least one translocation in any patient sample. Similarly, for the structural variant data, genes were included on the list if they

had at least one structural variant in any patient sample. For each mutation data set, a background data set made up of genes with similar replication timing and expression was used. Expression was defined as the average expression from the mock, 160 min E2-treated and 1280 min E2-treated RNA-seq samples. Replication timing was determined from the mean wavelet smoothed RepliSeq signal across each gene body, using ENCODE MCF7 RepliSeq data (accession number wgEnco-deEH002247). We used custom python scripts to match each translocated gene to a similar counter-part (scripts: https://github.com/cimprichlab/stork_et_al_analysis). Briefly, the replication and expression data were studentized, and the Euclidean distance between each mutated and non-mutated gene was calculated. For every gene in the mutated set, we picked the closest gene by Euclidean distance that was not currently in the matched set. We then compared the Z-score of estrogen induction between the test and matched sets.

Simple somatic mutation data were obtained from the ICGC commons (*Nik-Zainal et al., 2016*). Using the previously-calculated z-scores for estrogen induction from the RNA-seq dataset, genes with z-score greater than 2 were categorized as induced, while genes with z-score less than 2 were categorized as non-induced. The raw numbers of simple somatic mutation events were counted over each gene in the RNA-seq dataset, and the counts were normalized to the length of the gene in kilo-bases as defined by the UCSC gene associated with the gene symbol. Genomic regions matched for replication timing and expression were calculated as described above. Data were log transformed before plotting.

Significance was assessed by a two-tailed bootstrap of the median, where the data were re-sampled into sets with the same size as the test and matched sets 1,000,000 times, and the median was calculated for both sets. The p-value was calculated as the fraction of instances that the absolute value of the difference in medians between the resampled test and matched set was greater than the absolute value of the difference in median between the true test and matched sets.

## Proximity ligation assay

For the proximity ligation assay (PLA), cells were pre-extracted with cold 0.5% NP-40 for 3 min on ice. Cells were then fixed with 4% PFA/PBS for 15 min, washed 3 times with 1X PBS and blocked for 1 hr at RT with 2% BSA/PBS. Cells were then incubated with primary antibody overnight at 4°C (1:200 mouse S9.6 antibody alone; 1:500 rabbit P-H2AX alone; or 1:200 mouse S9.6 with 1:500 rabbit P-H2AX). Cells were washed 3 times in 1X PBS and incubated in a pre-mixed solution of PLA probe anti-mouse minus and PLA probe anti-rabbit plus (Sigma) for 1 hr at 37°C. The Duolink In Situ Detection Reagents (Green) were then used to perform the PLA reaction according to the manufacturer's instructions. Slides were mounted in Duolink In Situ Mounting Medium with DAPI and imaged on a Zeiss Axioscope at 40X. The number of PLA foci was quantified using Image J.

## S9.6 co-IP

Cells were trypsinized, washed in 1X PBS, and resuspended in 25 mL of 1X PBS. Cells were cross-linked in 1% formaldehyde (Pierce), quenched with 0.125 M glycine, and washed 2 times in 1X PBS containing protease inhibitor (PI; Roche). The cells were lysed in lysis buffer (50 mM HEPES pH 7.9, 140 mM NaCl, 1 mM EDTA, 10% glycerol, 0.5% NP-40, 0.25% Triton X-100) with PI and chromatin was sonicated in shearing buffer (0.1%SDS, 1 mM EDTA, 10 mM Tris pH 8.1) on a Covaris to an average size of 1 kb. Washed Protein A/G sepharose beads (Pierce) were used to pre-clear chromatin for 2 hr. 10 μg of chromatin and 20 μg of S9.6 antibody or 20 μg mouse IgG were incubated overnight at 4°C. Pre-washed protein A/G sepharose beads were then added to chromatin/antibody mixture for 2 hr. After washing three times in binding buffer (10 mM NaPO4 pH 7, 140 mM NaCl, 0.05% Triton X-100), bound beads were boiled in 30 μL 5X sample buffer and loaded on a 4–20% gradient gel (Biorad).

## Replicates and statistical analysis (for sequencing see relevant methods)

In this study, biological replicates indicate replicates of the experiment performed with separate passages of a cell line and independently prepared reagent mixes, most often at different times. Technical replicates indicate distinct measurements from distinct wells or coverslips from the same biological experiment, processed at the same time. Box and whiskers in all graphs represent 25–75 and 10–90 percentiles, respectively. The line represents the median value. All box and whisker plots

show representative experiments. Prism v6 (GraphPad Software) was used to perform student's t-test and non-parametric Mann-Whitney rank sum t-test, where noted. Error bars represent standard deviation (SD) unless otherwise noted as standard error of the mean (SEM). $*p<0.05$, $**p<0.01$, $***p<0.001$, and $****p<0.0001$.

## Acknowledgements

We thank the members of the Cimprich lab for critical input, especially Stephan Hamperl, who helped with S9.6 antibody purification. We thank Joanna Wysocka for helpful feedback on the manuscript. CTS was supported by an NSF GRFP and the NIH (T32GM007276). This work was supported by a Komen Foundation grant to KAC (IIR 12222368), NIH grants to KAC (RO1 GM119334, RO1 GM100489), and an NIH grant to FC (R01 GM094299). The authors declare no conflicts of interest.

## Additional information

### Funding

| Funder | Grant reference number | Author |
|---|---|---|
| Susan G. Komen | IIR 12222368 | Karlene A Cimprich |
| National Institutes of Health | R01 GM119334 | Karlene A Cimprich |
| National Institutes of Health | R01 GM100489 | Karlene A Cimprich |
| National Institutes of Health | R01 GM094299 | Frédéric Chédin |
| National Science Foundation | Graduate Research Fellowship | Caroline Townsend Stork |
| National Institutes of Health | Training Grant T32GM007276 | Caroline Townsend Stork |

The funders had no role in study design, data collection and interpretation, or the decision to submit the work for publication.

### Author contributions

CTS, Conception and design, Acquisition of data, Analysis and interpretation of data, Drafting or revising the article; MB, TS, Analysis and interpretation of data, Drafting or revising the article; MPC, Acquisition of data, Drafting or revising the article; JS, Conception and design, Drafting or revising the article; LAS, Taught CTS how to perform DRIP-seq, Conception and design, Drafting or revising the article; FC, Taught CTS. how to perform DRIP-seq, Conception and design, Drafting or revising the article; KAC, Conception and design, Analysis and interpretation of data, Drafting or revising the article

### Author ORCIDs

Karlene A Cimprich, http://orcid.org/0000-0002-1937-2969

## Additional files

### Supplementary files

• Supplementary file 1. DRIP-qPCR primers.

### Major datasets

The following dataset was generated:

| Author(s) | Year | Dataset title | Dataset URL | Database, license, and accessibility information |
|---|---|---|---|---|
| Stork CT, Bocek MJ, Crossley MP, Sollier J, Sanz LA, Chédin F, Swigut T, Cimprich K | 2016 | Genome-wide DRIP-seq in E2 stimulated MCF7 cells | http://www.ncbi.nlm.nih.gov/geo/query/acc.cgi?acc=GSE81851 | Publicly available at the NCBI Gene Expression Omnibus (accession no: GSE81851) |

The following previously published datasets were used:

| Author(s) | Year | Dataset title | Dataset URL | Database, license, and accessibility information |
|---|---|---|---|---|
| Hah N, Danko CG, Core L, Watterfall JJ, Siepel A, Lis JT, Kraus WL | 2011 | Global Analysis of the Immediate Transcriptional Effects of Estrogen Signaling Reveals a Rapid, Extensive, and Transient Response | https://www.ncbi.nlm.nih.gov/geo/query/acc.cgi?acc=GSE27463 | Publicly available at the NCBI Gene Expression Omnibus (accession no: GSE27463) |
| Honkela A, Peltonen J, Topa H, Charapitsa I, Matarese F, Grote K, Stunnenberg HG, Reid G, Lawrence ND, Rattray M | 2014 | Genome-wide modelling of transcription kinetics reveals patterns of RNA processing delays | https://www.ncbi.nlm.nih.gov/geo/query/acc.cgi?acc=GSE62789 | Publicly available at the NCBI Gene Expression Omnibus (accession no: GSE62789) |
| Chambers VS, Marsico G, Boutell JM, Di Antonio M, Smith GP, Balasubramanian S | 2015 | High-throughput sequencing of DNA G-quadruplex structures in the human genome | https://www.ncbi.nlm.nih.gov/geo/query/acc.cgi?acc=GSE63874 | Publicly available at the NCBI Gene Expression Omnibus (accession no: GSE63874) |
| Nik-Zainal S, Davies H, Staaf J, Ramakrishna M, Glodzik D, Zou X, Martincorena I, Alexandrov LB, Martin S, Wedge DC | 2016 | Landscape of somatic mutations in 560 breast cancer whole-genome sequences | https://www.ebi.ac.uk/ega/studies/EGAS00001001178 | Publicly available at the European Genome-phenome Archive (accession no. EGAS00001001178) |

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
