## [Decision Letter]

Thank you for submitting your article "Estrogen-induced DNA damage is caused by co-transcriptional R-loops" for consideration by *eLife*. Your article has been reviewed by three peer reviewers, and the evaluation has been overseen by Andrés Aguilera as the Reviewing Editor and James Manley as the Senior Editor. The following individuals involved in review of your submission have agreed to reveal their identity: Andre Nussenzweig (Reviewer #1).

The reviewers have discussed the reviews with one another and the Reviewing Editor has drafted this decision to help you prepare a revised submission.

Summary:

The authors observe that MCF-7 cells display increases in DNA damage after exposure to prolonged periods of estrogen. This DNA damage is demonstrated by the appearance of P-H2AX, 53BP1 foci, and increased tail moment by neutral comet assay, suggesting these are DNA DSBs. The DNA damage is dependent on both transcription and DNA replication. The significance of the manuscript centers around the idea that DNA DSBs are dependent on R-loops formed by estrogen receptor-mediated transcription, which may saturate co-transcriptional machinery. The work describes that R-loops are formed after estrogen induction, showing convincingly that TC-NER machinery is required for P-H2AX and DSB formation.

Essential revisions:

The demonstration that H2AX is phosphorylated near R-loops using a proximity-ligation assay (PLA) does not seem to be entirely convincing. The PLA signal is mostly restricted to one focus, which is quite reminiscent of the single DNA damage response focus that can be detected in these cells with 53BP1 or pH2AX antibodies. It would be important therefore to try to complement with new information probing the proximity of R loops to DNA breaks. Since it seems that using the PLA assay most of the time only one S9.6-P-H2AX foci is observed, this could mean that not all P-H2AX foci originate (directly) from a R-loop, or that H2AX is phosphorylated at different genomic loci than where R-loops are formed. This can be tested by P-H2AX ChIP-qPCR after E2 induction on the E2 responsive genes depicted in Figure 2 upon E2 induction. Alternatively, or additionally, you could explore whether genomic rearrangements occurring in ER positive tumors are enriched at ER target sites. The recent Nature paper (Nik-Zainal S et al. 2016) cataloging all genomic rearrangements in more than 500 breast cancers allows this analysis.

---

## [Author Response]

Essential revisions:

The demonstration that H2AX is phosphorylated near R-loops using a proximity-ligation assay (PLA) does not seem to be entirely convincing. The PLA signal is mostly restricted to one focus, which is quite reminiscent of the single DNA damage response focus that can be detected in these cells with 53BP1 or pH2AX antibodies. It would be important therefore to try to complement with new information probing the proximity of R loops to DNA breaks. Since it seems that using the PLA assay most of the time only one S9.6-P-H2AX foci is observed, this could mean that not all P-H2AX foci originate (directly) from a R-loop, or that H2AX is phosphorylated at different genomic loci than where R-loops are formed.

While there is a clear increase in PLA signal between the S9.6 antibody and the P-H2AX antibody following E2 addition, the reviewers correctly note that the PLA signal is often limited to 1-2 foci per cell. Although this could indicate that the DNA damage and R-loops do not coincide and breaks arise indirectly from R-loops, there are other interpretations as well. For example, one possibility, as the reviewers suggest, is that the H2AX is phosphorylated at loci far enough downstream of where the R-loop forms such that the distance falls outside of a detectable interaction for the PLA assay. As RNA-DNA hybrids are not expected to bind nucleosomes (Dunn and Griffith, 1980), this is quite feasible and may limit the sensitivity of this assay. We have included a sentence in the main text of the Results where we address this possibility: “The relatively small number of detectable interactions between the RNA-DNA hybrid and P-H2AX antibodies upon E2 addition may indicate that the majority of DNA damage occurs at a distance from the R-loop that is outside the limits of PLA detection (<30-40 nM)” Nevertheless, we do recognize that this leads to questions about the proximity of DNA damage and R-loops, and we try to address this issue as explained below.

This can be tested by P-H2AX ChIP-qPCR after E2 induction on the E2 responsive genes depicted in Figure 2 upon E2 induction. Alternatively, or additionally, you could explore whether genomic rearrangements occurring in ER positive tumors are enriched at ER target sites. The recent Nature paper (Nik-Zainal S et al. 2016) cataloging all genomic rearrangements in more than 500 breast cancers allows this analysis.

We thank the reviewers for the suggestions and have performed the recommended computational experiment to determine if breast cancer rearrangements are enriched in regions that are ER target sites. For this analysis, we defined ER target sites as regions of the genome that are transcriptionally responsive to estrogen (E2) as measured by RNA-seq, since ER binding can occur at sites distant from the transcriptional response. As suggested, we utilized the rearrangement data from the whole-genome sequences of 560 breast cancers from Nik-Zainal et al., 2016. We took into account mutations from both ER-positive and ER-negative breast tumors because precursor cells can express high levels of the ER at some point during development (Allred et al., 2004) and therefore could be affected by E2 signaling. Notably, we see that E2-responsive regions are significantly enriched among both breast cancer structural variants and translocations (p<1.0 x 10^-6^). Interestingly, not all mutation types are enriched in E2-responsive genes, as we do not see an enrichment of the simple somatic mutations such as substitutions and indels in E2-responsive genes. We have now included these data as a new Figure 3 and describe the results in the subsection “Breast cancer rearrangements are enriched in E2-responsive genes”. To ensure that the enrichment we observe is not biased by the gene expression level or replication timing, the latter of which significantly affects mutation frequency (Stamatoyannopoulos et al., 2009), we used background models that match the expression level and replication timing of the genes within each mutation dataset, as shown in Figure 3—figure supplement 1–Figure 3—figure supplement 3. We believe that this data strongly supports the notion that E2-responsive regions of the genome are more frequently rearranged in breast cancer.

We have also expended considerable effort to map P-H2AX in response to E2 by ChIP. However, we have not been able to detect a clear, reproducible increase in ChIP signal upon E2 treatment relative to signal from regions of the genome that do not contain R-loops. Given the potential for megabase-scale spreading of P-H2AX, it is challenging to determine if the increase in signal in non-R-loop regions is due to P-H2AX spreading or poor signal-to-noise. It is also possible we have not chosen the appropriate sites, since we do not expect that double-strand breaks will form at each R-loop in each cell. We have tried performing genome-wide P-H2AX ChIP-seq to help differentiate between these possibilities, but did not detect significant enrichment. We believe that a more sensitive method of double-strand break mapping may be required to definitely map site-specific damage, as the lack of enrichment is an issue with the P-H2AX ChIP approach. Even under ideal conditions when a double-strand break is induced in all cells with the restriction endonuclease AsiSI (Aymard et al., 2014; Iacovoni et al., 2010), we observe enrichment for P-H2AX is 1.8 fold at best.

References

Allred, D.C., Brown, P., and Medina, D. (2004). The origins of estrogen receptor alphapositive and estrogen receptor alpha-negative human breast cancer. Breast Cancer Res 6, 240-245.

Aymard, F., Bugler, B., Schmidt, C.K., Guillou, E., Caron, P., Briois, S., Iacovoni, J.S., Daburon, V., Miller, K.M., Jackson, S.P., et al. (2014). Transcriptionally active chromatin recruits homologous recombination at DNA double-strand breaks. Nat Struct Mol Biol 21, 366-374.

Dunn, K., and Griffith, J.D. (1980). The presence of RNA in a double helix inhibits its interaction with histone protein. Nucleic Acids Res 8, 555-566.

Iacovoni, J.S., Caron, P., Lassadi, I., Nicolas, E., Massip, L., Trouche, D., and Legube, G. (2010). High-resolution profiling of gammaH2AX around DNA double strand breaks in the mammalian genome. EMBO J 29, 1446-1457.

Nik-Zainal, S., Davies, H., Staaf, J., Ramakrishna, M., Glodzik, D., Zou, X., Martincorena, I., Alexandrov, L.B., Martin, S., Wedge, D.C., et al. (2016). Landscape of somatic mutations in 560 breast cancer whole-genome sequences. Nature 534, 47-54.

Stamatoyannopoulos, J.A., Adzhubei, I., Thurman, R.E., Kryukov, G.V., Mirkin, S.M., and Sunyaev, S.R. (2009). Human mutation rate associated with DNA replication timing. Nat Genet 41, 393-395.